# Membrane Status and Reliability of Intrapartum Transperineal Ultrasound in Cervical Dilatation Assessment

**DOI:** 10.3390/healthcare13111322

**Published:** 2025-06-02

**Authors:** George-Alexandru Roșu, Dan-Bogdan Navolan, Adrian Neacșu, Ștefan-Florentin Semeș, Crîngu-Antoniu Ionescu

**Affiliations:** 1“Carol Davila” University of Medicine and Pharmacy-Doctoral School (IODS), 050474 Bucharest, Romania; 2Obstetrics & Gynecology Department, “Sfântul Pantelimon” Clinical Emergency Hospital, “Carol Davila” University of Medicine and Pharmacy, 021659 Bucharest, Romania; antoniuginec@yahoo.com; 3Obstetrics & Gynecology Department, “Victor Babeș” University of Medicine and Pharmacy, 300041 Timișoara, Romania; navolan@yahoo.com; 4Obstetrics & Gynecology Department, “Bucur” Maternity, “Carol Davila” University of Medicine and Pharmacy, 040294 Bucharest, Romania; adrianneacsu2006@yahoo.com; 5Obstetrics & Gynecology Department, “Sfântul Pantelimon” Clinical Emergency Hospital, 021659 București, Romania; semes.stefan5@gmail.com

**Keywords:** dilatation, membranes, intrapartum ultrasound

## Abstract

**Background and Objectives:** Labor progression evaluation through repeated vaginal examinations remains the primary method of monitoring in delivery rooms globally. Transperineal intrapartum ultrasound has been shown to be reliable for assessing cervical dilatation, with substantial concordance with digital vaginal examinations. However, none of the analyzed studies investigated the influence of membrane integrity on ultrasound measurements. This study assessed the impact of membrane status on cervical dilatation evaluation via transperineal ultrasound compared to clinical examination, and the extent of agreement based on dilatation level and membrane status. **Methods**: A nine-month longitudinal observational study was conducted in the Obstetrics and Gynecology Clinic of “Sfântul Pantelimon” Clinical Emergency Hospital (Bucharest, Romania). Patients underwent two clinical examinations and two transperineal ultrasound measurements, one at a dilatation less than 8 cm and the other at a dilatation closer to full dilatation (above 8 cm). Agreement between clinical and ultrasound measurements was analyzed based on membrane integrity and dilatation level. **Results**: In total, 239 patients were included, and 478 cervical dilatation measurements were obtained. Only the 7–8 cm subgroup exhibited statistically significant differences in accuracy between patients with intact and ruptured membranes. The Pearson correlation results for membrane status were 0.87 (p-value < 0.001) for intact membranes and 0.91 (p-value < 0.001) for ruptured membranes. Both groups show a strong positive correlation, suggesting that ultrasound and clinical measurements tend to increase simultaneously, regardless of membrane status. **Conclusions**: Transperineal ultrasound is useful for labor monitoring, but its accuracy decreases significantly in advanced labor, especially beyond 8 cm dilatation and in cases with ruptured membranes.

## 1. Introduction

Labor progression evaluation through repeated vaginal examinations is still the main method of monitoring in delivery rooms around the globe because it is a cheap and non-invasive method. But this way of labor monitoring has a certain degree of inaccuracy and is a maneuver that creates discomfort for the patient. The publication of guidelines related to ultrasound in labor in 2018 and 2022 [1,2] did not significantly increase the use of intrapartum ultrasound monitoring. The search to identify an objective method to evaluate cervical dilatation during labor, a method that could increase the accuracy of clinical evaluation, has its origins in a publication by Frans Kok et al. from 1976, in which an ultrasonic cervimeter was proposed for this specific task, a machine for which the study reported an average error of 3 mm for dilatation evaluation from 2 to 10 cm [3]. This technique was further studied by Moss et al.; the study was published showing promising results for continuous recording of cervical dilatation using the ultrasonic cervimeter imagined by Kok and colleagues 2 years before [4]. In a review published by van Dessel et al. in 1991, in which his team analyzed the various techniques of dilatation measurement, the ultrasound cerimetry, the internal ultrasound machine, and the technique of measurement described and studied by Kok in 1976, and later by Moss in 1978, was deemed “a useful research tool for the study of cervical response to the uterine contractions during labor” and it was concluded that “For clinical obstetric purposes, however, digital assessment of cervical dilatation seems sufficient” [5]. After 22 years Hassan and Eggebo et al. published a paper that analyzed 2D transperineal ultrasound during labor for cervical dilatation assessment and concluded that this method is reliable and has a close agreement with digital vaginal examination [6]. Several studies have shown that transperineal ultrasound is a reliable method for assessing cervical dilatation, showing good agreement with digital vaginal examinations, thus being recommended as a complement to traditional clinical examinations, particularly in special situations such as preterm premature membrane rupture, women in the first stage of labor, and women with suspicion of preterm labor [7,8,9,10,11,12]. In none of these studies did we find a focused comparison evaluating the accuracy and reliability of transperineal ultrasound during labor for cervical dilatation measurement between ruptured and unruptured membranes, nor the influence of cervical dilatation level on the accuracy of measurement compared with digital vaginal examination. The aim of this study was to evaluate the impact of membrane status on cervical dilatation evaluation through transperineal ultrasound compared with clinical examination and to determine the degree of agreement between these two methods according to dilatation level and membrane status. This study was conceived and developed in light of the increasing use of ultrasound examination during labor in the past 5–10 years around the globe with the hope of obtaining more data that would support the use of this technique to increase the objectivity and quality of labor monitoring in modern obstetric practice.

## 2. Materials and Methods

This prospective longitudinal observational study was conducted in the Obstetrics & Gynecology Clinic of “Sfântul Pantelimon” Clinical Emergency Hospital between 1 April 2022 and 31 December 2022. This study was approved by the Ethics Committee of the “Sfântul Pantelimon” Clinical Emergency Hospital (Decision no. 7/26 January 2022). All women provided written informed consent for study participation. The inclusion criteria for this study were singleton pregnancy, cephalic presentation, and spontaneous labor. The exclusion criteria were multiple pregnancy, breech presentation, induced labor, and previous cesarean section. Patients without good-quality images of the dilatation were also excluded from this study. In total, 239 patients were included in this study based on these criteria. Digital vaginal examinations were performed twice by an attending physician or on-call doctor for this study. All physicians who performed the vaginal examinations had at least 5 years of experience. In total, 8 experienced physicians performed the vaginal examinations during this study. Immediately after performing the vaginal examination, between contractions, transperineal ultrasound was performed by other physicians with at least 5 years of experience performing ultrasound examinations, and who received prior training for performing transperineal ultrasound in labor. Five experienced and specifically trained physicians performed the ultrasound examinations in this study. The physician performing the transperineal ultrasound was blinded to the physician performing the clinical examination. The ultrasound machine was a Voluson E10 BT20 (GE Healthcare Austria GmbH & Co OG, Tiefenbach, Austria), and the C1-6 convex array probe was used. The probe was covered with a sterile glove and ultrasound gel. The transducer was disinfected after each use. The technique of examination was the one described in the original article by Hassan et al. in 2013 [6] and was replicated in the following years in other publications [8,11]. For each patient we performed 2 measurements through transperineal ultrasound: one at a dilatation under 8 cm and another closer to full dilatation (greater than 8 cm). For each measurement, the physician measured the anteroposterior and transverse diameters and calculated the mean diameter. The anteroposterior diameter of the dilatation was defined and measured by placing the calipers from the inside of the anterior limit of the dilatation up to the most distant and visible posterior limit of the dilatation. The transverse diameter was defined and measured by placing the calipers from the inside of one of the lateral limits of the dilatation up to the most lateral and visible limit of the dilatation. The mean diameter of the dilatation was obtained by summing the measured anteroposterior and transverse diameters and dividing the sum by two.

### Statistical Analysis

Descriptive analyses were performed using Microsoft Excel for Mac Version 16.95.4. Inferential statistical analyses were conducted utilizing SPSS V.28 (IBM). The agreement between clinical cervical dilatation and ultrasound measurements depending on membrane integrity and dilatation level was evaluated using multiple statistical methods. The degree of agreement was first assessed using Cohen’s kappa coefficient, treating clinical and ultrasound mean diameters as categorical variables. To evaluate the strength of association, the Pearson correlation coefficient was calculated for both the overall cohort and stratified by membrane status (intact vs. ruptured). Linear regression analysis was used to model the relationship between clinical and ultrasound measurements. Furthermore, Bland–Altman plots were constructed to illustrate the agreement between clinical examination and both the mean and transverse ultrasound diameters, providing mean differences and 95% limits of agreement. These analyses were also stratified by membrane status to explore potential differential agreement based on membrane integrity. All results underwent rigorous review for accuracy and consistency prior to inclusion.

## 3. Results

In total, 239 patients were included in this study during the nine-month interval based on the inclusion and exclusion criteria. For each patient, we included in the analysis two clinical examinations that were compared for accuracy and agreement with two transperineal ultrasound examinations. The total number of clinical examination and ultrasound examination pairs was 478.

### 3.1. Population Characteristics

The mean age of the patients included in the study was 26 years old, with the majority of patients coming from rural areas (132 out of 239 patients, 55.23%) (Table 1). The patients had their pregnancy followed up in 62.34% of cases. The average gestational age at admission was 38 weeks, varying between 30 weeks and 41 weeks of gestation. At admission, 50.21% of patients had intact membranes (120 out of 239 patients). In cases admitted with ruptured membranes, only 4.18% of the patients presented with meconium-stained fluid. The average hospital stay was 5 days, the duration varied between 1 day and 34 days. Most patients delivered the same day that they were admitted (91.83%).

The mean length of labor was 281 min, with a minimum duration of 70 min and the longest labor duration being 870 min (14 h and 30 min). The mean number of vaginal examinations per labor was four, varying between two and six. The average number of vaginal examinations per hour was one, with a maximum of two exams per hour.

Analyzing the fetal and neonatal characteristics (Table 2), the average neonatal weight was 3057 g, varying between 1430 g and 4140 g. Comparing the neonatal weight with the estimate fetal weight evaluated by ultrasonography during labor, we observed a mean difference between the estimated fetal weight and neonatal weight of 197 g (calculated in the absolute values group), with a maximum underestimation of 730 g and a maximum overestimation of 632 g, with a median difference of 20 g when we evaluated the median in the integer value group. Most of the neonates were female (53.97%—129 out of 239 neonates). The average Apgar score at 1 min was 9, varying between 1 and 9.

### 3.2. Ultrasound Measurements

The ultrasound measurements were performed as presented in the Materials and Methods section, based on the descriptions of previous studies regarding the technique that is necessary to obtain cervical dilatation measurements through transperineal ultrasound in labor. Out of 478 measurements, 349 were performed on ruptured membranes (73.01%), and the rest were performed on intact membranes. The measurements performed on intact membranes were easier to perform due to the better visualization of the margins of the dilatation, and the placement of the calipers was better (as can be seen in Figure 1a,c. When performed on ruptured membranes, the margins of the cervical dilatation were seen with slightly more difficulty, with the examination taking more time (Figure 1b). Additionally, when the fetal head was engaged and descended, the examinations became more difficult, with a lower visualization score (Figure 1d). For each evaluation, we measured the anteroposterior and transverse diameters of the dilatations, as shown in Figure 1. The mean of the two diameters was calculated after the examination. All measurements were analyzed and compared with the mean diameter of the clinical dilatation. From each dilatation range or level, we considered the mean of the interval as the value of the clinical dilatation, eg., for 3–4 cm clinical dilatation, we considered 35 mm the clinical dilatation value with which we performed the statistical analysis.

### 3.3. Membrane Integrity Influence on Ultrasound Parameter Measurement Accuracy

Among the 478 ultrasound measurements analyzed across various cervical dilatation subgroups, only the 7–8 cm subgroup showed statistically significant differences in accuracy between patients with intact versus ruptured membranes (Table 3). Specifically, for mean diameter (*p* = 0.0224) and transverse diameter (*p* = 0.0109), ultrasound measurement error was significantly influenced by membrane status. For all other dilatation levels and ultrasound methods, including Anteroposterior measurements, no significant differences were found between membrane status groups (*p* ≥ 0.05), indicating the overall consistency of ultrasound accuracy across membrane conditions, except at this transitional stage of labor.

In Table 4, we analyze the most accurate diameter measurement by transperineal ultrasound for each clinical dilatation range, considering the mean difference between the measurements performed on intact membranes versus ruptured membrane measurements. The mean diameter is the most accurate method across all the stages below 8 cm dilatation. Closer to full dilatation, the accuracy of the mean diameter is lower compared with the transverse diameter.

The analysis of the difference between clinical dilatation evaluation and mean diameter of the dilatation calculated by ultrasound measurements and stratified by clinical dilatation range and membranes status can be seen in the violin plot presented in Figure 2. This analysis reveals that ultrasound tends to underestimate dilatation more in cases with ruptured membranes, especially as dilatation progresses. The same trend was observed when we performed a sub-analysis on the transverse and anteroposterior diameters compared with the mean clinical dilatation diameter. In Figure 3, we analyze the mean difference trend between clinical dilatation evaluation and mean diameter of the dilatation calculated by ultrasound measurement, stratified by clinically estimated cervical dilatation range and membrane integrity. A consistent underestimation is noted for ruptured membranes, with the largest discrepancy at full dilatation. The intact membrane group showed smaller differences.

### 3.4. Membrane Status Influence on the Agreement Between Clinical and Ultrasound Examinations

We evaluated the agreement between the clinical evaluations and ultrasound measurements by Cohen Kappa coefficient, Pearson correlation, linear regression, and Bland–Altman analysis.

#### 3.4.1. Cohen Kappa Coefficient

The Cohen’s Kappa value between the clinical examination and the ultrasound mean diameter measurements is approximately 0.052. This indicates slight agreement beyond chance, suggesting that the two methods do not strongly align when treated as categorical (integer-based) measurements. The Cohen’s Kappa results by membrane status are the following: for intact membranes, κ ≈ 0.045, and for ruptured membranes, κ ≈ 0.052. Both indicate slight agreement between clinical and ultrasound mean diameter measurements.

#### 3.4.2. Pearson Correlation

The Pearson correlation results by membrane status are as follows: for intact membranes, the correlation was 0.87 (*p*-value < 0.001) with a 95% CI [0.821,0.907], and for ruptured membranes, the correlation was 0.91 (*p*-value < 0.001) with a 95% CI [0.890–0.926].

#### 3.4.3. Linear Regression Analysis

A linear regression analysis was performed on all cases and for the two subgroups, intact membranes and ruptured membranes, as follows (Figure 4 and Figure 5):Overall (all cases)R-squared: 0.856 (very strong), *p*-value: <0.001 (highly significant);Regression equation: Ultrasound Mean Diameter ≈ 13.94 + 0.67 × Clinical Dilatation.Intact Membranes SubgroupR-squared: 0.751, *p*-value: <0.001;Regression equation: Ultrasound Mean Diameter ≈ 10.37 + 0.75 × Clinical Dilatation.Ruptured Membranes SubgroupR-squared: 0.832, *p*-value: <0.001;Regression equation: Ultrasound Mean Diameter ≈ 14.41 + 0.67 × Clinical Dilatation.

The intact membrane group plot (Figure 4) shows a clear upward trend, though with slightly more spread in the data.

The ruptured membrane plot (Figure 5) has a tighter fit around the regression line, reflecting the higher R-squared value.

#### 3.4.4. Bland–Altman Analysis

The Bland–Altman statistics comparing clinical dilatation evaluation and mean ultrasound diameter of the dilatation calculated after ultrasound measurements and by membrane status (Figure 6) show the following:For intact membranes, the differences between clinical dilatation evaluation and mean diameter of the dilatation calculated by ultrasound measurements show slightly more variability, and the bias (mean difference) may be a bit more negative (Mean Difference: 1.83 mm, Limits of Agreement: Upper: +20.20 mm, Lower: −16.54 mm.For ruptured membranes, the differences between clinical dilatation evaluation and mean diameter of the dilatation calculated by ultrasound measurements cluster more tightly around the mean, indicating more consistent agreement (Mean Difference: 13.76 mm, Limits of Agreement: Upper: +37.10 mm, Lower: −9.58 mm).

The Bland–Altman statistics comparing clinical dilatation evaluation and transverse ultrasound diameter of the dilatation measured by ultrasound and by membrane status (Figure 6) have the following results:For intact membranes, the transverse diameter closely matches clinical measurements on average (mean difference is near zero), though the spread is wide (Mean Difference: 0.45 mm, Limits of Agreement: Upper: +22.15 mm, Lower: −21.25 mm).For ruptured membranes, clinical measurements are much higher than transverse ultrasound measurements on average, with a large and consistent bias (Mean Difference: 13.18 mm, Limits of Agreement: Upper: +41.22 mm, Lower: −14.86 mm).

The layout in Figure 6 clearly shows that the transverse diameter has less bias in intact membrane measurements, but with larger variability, and that the mean diameter has more consistent agreement overall, especially in ruptured membrane measurements.

## 4. Discussion

Similarly to previous studies that have shown that transperineal ultrasound is a reliable method for assessing cervical dilatation, showing good agreement with digital vaginal examinations and a strong positive correlation, and thus being recommended as a feasible method of labor monitoring complementary to traditional clinical examinations [7,8,9,10,11,12,13,14,15], our study has comparable results regarding the degree of agreement and correlation of these two methods. In addition to these aspects, our study provides data related to the influence of membrane integrity on ultrasound measurement, being the first study to analyze this aspect in depth. There are also studies that conclude that the sonographic approach for dilatation evaluation is at least as accurate as the clinical examination and include it in a sonopartogram, a modern partogram in which the acquisition of labor progression-related data was more successful compared with the conventional partogram, thus recommending it for more accurate labor progression monitoring [9,16,17]. The current study has also used a modified sonopartogram, derived from the ones previously mentioned, to acquire all the data related to labor progression monitoring, for at least two occasions during labor, seeing the same increase in data acquisition and data quality. More studies are necessary to evaluate the feasibility of the sonopartogram and to establish a more structured sonopartogram form with specific moments during labor for transperineal ultrasound evaluation, and to establish which parameters should be monitored for each various moment, before including the sonopartogram in the routine labor monitoring.

One aspect for which it is difficult to find an explanation is the fact that membrane rupture may contribute to the underestimation of dilatation in the 7–8 cm range by mean and transverse diameter measured by transperineal ultrasound, especially in cases of ruptured membranes, and not in the other clinical dilatation ranges and membrane statuses. The dilatation limit that separated the lower dilatations ranges from the dilatation ranges closer to full dilatation. Also, as a general trend, this analysis reveals that ultrasound tends to underestimate dilatation more in cases with ruptured membranes, especially as dilatation progresses, with this study being the first to conclude this, and previous studies not performing this subgroup analysis.

The clinical dilatation evaluation is usually performed by the evaluation of the transverse diameter and rarely by evaluating the anteroposterior diameter. We compared this subjective measurement, although performed by practitioners with experience in evaluating cervical dilatation, with the mean diameter of the dilatation calculated from ultrasound measurement; for this reason, the mean difference between these dilatations could have a larger interval of agreement. This could also explain the fact that at larger dilatations, greater than 8 cm, the transverse diameter of the dilatation presented a higher accuracy compared with the mean diameter, a different result in comparison to the result of the study conducted by Usman et al. in 2018 and that of Hassan et al. from 2013 [6,18]. This discordance between these results in our study and from the literature could be explained by the measurement technique and the visibility of the dilatation margins considered important to measure or not, with this last aspect being dependent on the ultrasound machine and the resolution of the probe, but also on the experience of the physician.

Based on the Pearson correlation, both groups (intact and ruptured membranes) show a strong positive correlation, indicating that ultrasound and clinical measurements tended to increase together, regardless of membrane status. When evaluated by linear regression, all models showed a strong and significant linear relationship. These aspects could be used to recommend the use of transperineal ultrasound during labor to evaluate smaller dilatations (under 8 cm), to reduce the number of digital vaginal examinations, and the discomfort associated with it, as Wiafe et al. (2016) and Mohaghegh et al. (2021) concluded in their metanalyses [8,10]. The acceptability and tolerability of ultrasound examination compared with the digital vaginal examination for the evaluation of cervical dilatation has been found to be higher in a number of studies, in particular due to a reduced perception of pain and anxiety, especially in nulliparous women and in situations where a point-of-care ultrasound device is used at the bed of the patient [14,19,20,21,22,23]. Also, by reducing the number of vaginal examinations, we could reduce the risk of maternal and neonatal infections, as suggested by some studies [24,25,26], with this being another advantage of using transperineal ultrasound for labor progression monitoring instead of repeated vaginal examinations, especially in cases with membranes that have been ruptured for a longer time, as in the preterm premature rupture of the membrane or cases of membrane rupture at the beginning of labor. Taking into consideration these advantages, whenever possible, ultrasound examination for cervical dilatation should be used, especially at dilatations smaller than 8 cm.

The Bland–Altman analysis showed that clinical measurements tend to be higher than ultrasound in both intact and ruptured membranes, but especially so when membranes are ruptured. When we performed the Bland–Altman analysis for the transverse diameter measured by ultrasound, we concluded that this might be a rough substitute for the mean diameter, with less bias, but with larger variability, if this is acceptable. All these findings suggest that the rupture of the membrane may lead clinicians to systematically overestimate cervical dilatation, possibly due to changes in tissue palpability or examiner perception, but studies that investigate this specific cause of overestimation should be conducted.

This research, however, is subject to several limitations. The first limitation is represented by the usage of an ultrasound machine that was placed in one of the delivery rooms (in a specific place), where the patients were moved to have the ultrasound examinations performed, considering the fact that the patients were in labor. The second limitation was the time constraint, because the duration of this study was established at the beginning, and this limited the number of patients included. Another limitation of this study is the fact that we did not subdivide and compare the nulliparous women to the multiparous women when we discuss the accuracy of the evaluation at larger dilatations. Finally, the fourth limitation was represented by the fact that we did not consider the fetal head progression and its influence on measurements and visibility of cervical dilatation. All these limitations should be investigated in further research, possibly addressing them all together.

## 5. Conclusions

In light of the previous discussion and the limitations presented, intrapartum transperineal ultrasound can be a useful tool for labor monitoring and cervical dilatation evaluation due to high correlation with the clinical examination, especially in cases with intact membranes and dilatations lower that 8 cm, with its accuracy diminishing notably in advanced labor, particularly that beyond 8 cm dilatation and in cases with ruptured membranes. This technique of labor progression monitoring could be integrated into clinical practice to increase the objectivity of monitorization and to decrease the discomfort of pregnant women due to repeated clinical examinations.

## Figures and Tables

**Figure 1 healthcare-13-01322-f001:**
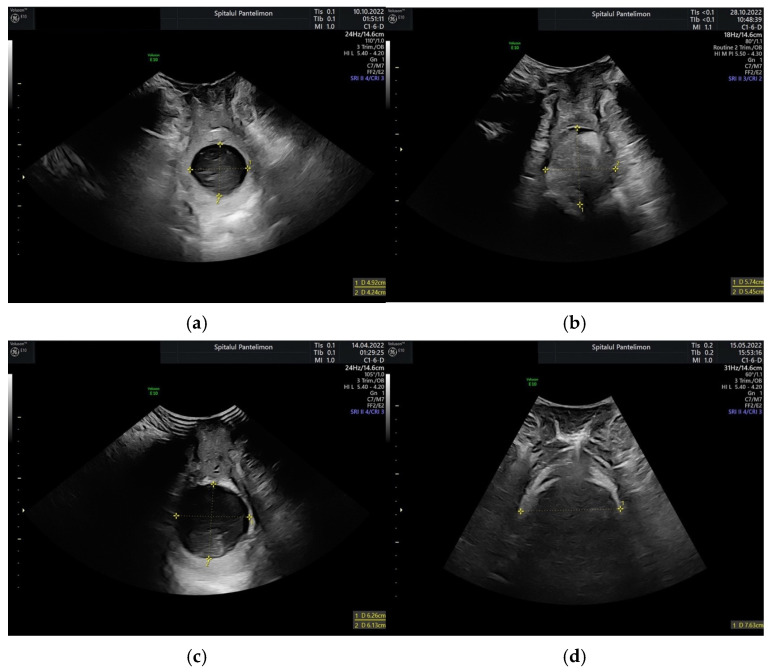
Transperineal intrapartum ultrasound images of various dilatation levels with intact or ruptured membranes: (**a**) 4–5 cm dilatation, intact membranes; (**b**) 5–6 cm dilatation, ruptured membranes; (**c**) 6–7 cm dilatation, intact membranes; (**d**) 7–8 cm dilatation, ruptured membranes.

**Figure 2 healthcare-13-01322-f002:**
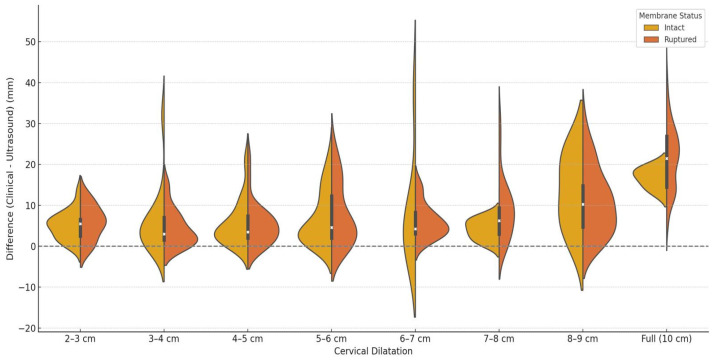
Difference between clinical dilatation evaluation and mean diameter of the dilatation calculated by ultrasound measurements, stratified by clinically estimated cervical dilatation range and membrane integrity.

**Figure 3 healthcare-13-01322-f003:**
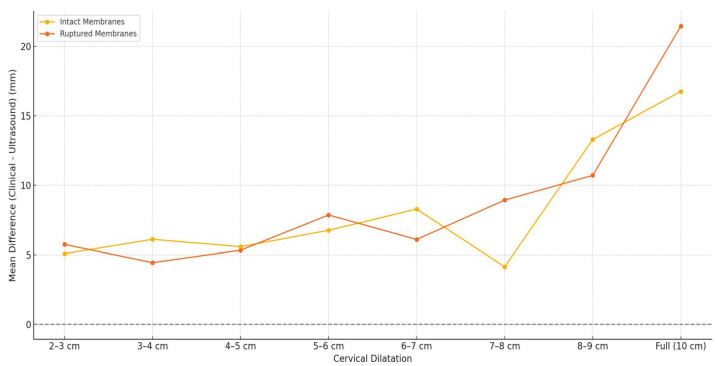
Mean difference between clinical dilatation evaluation and mean diameter of the dilatation calculated by ultrasound measurements, stratified by clinically estimated cervical dilatation range and membrane integrity—progressive underestimation in ruptured membranes as labor advances, reinforcing the pattern of greater discrepancy in ruptured membranes.

**Figure 4 healthcare-13-01322-f004:**
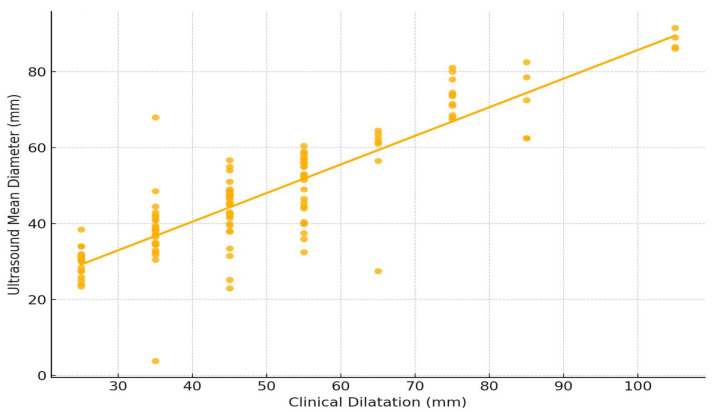
Linear regression graphic superimposed over Pearson correlation for clinical dilatation evaluation versus mean diameter of the dilatation calculated by ultrasound measurements for intact membrane subgroup.

**Figure 5 healthcare-13-01322-f005:**
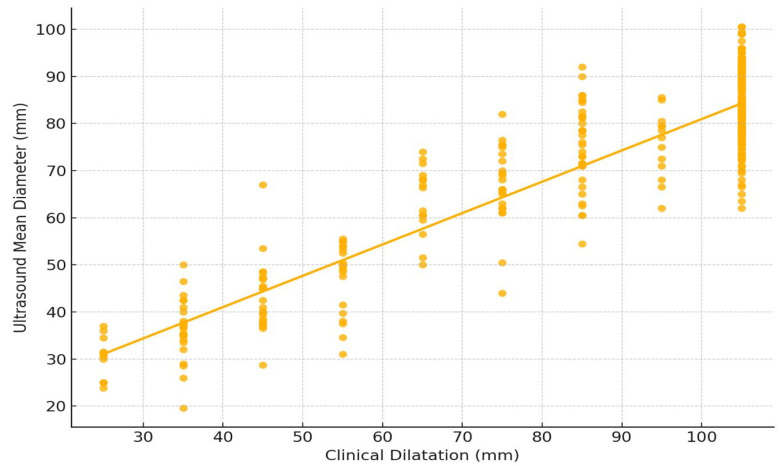
Linear regression graphic superimposed over Pearson correlation for clinical dilatation evaluation versus mean diameter of the dilatation calculated by ultrasound measurements for ruptured membrane subgroup.

**Figure 6 healthcare-13-01322-f006:**
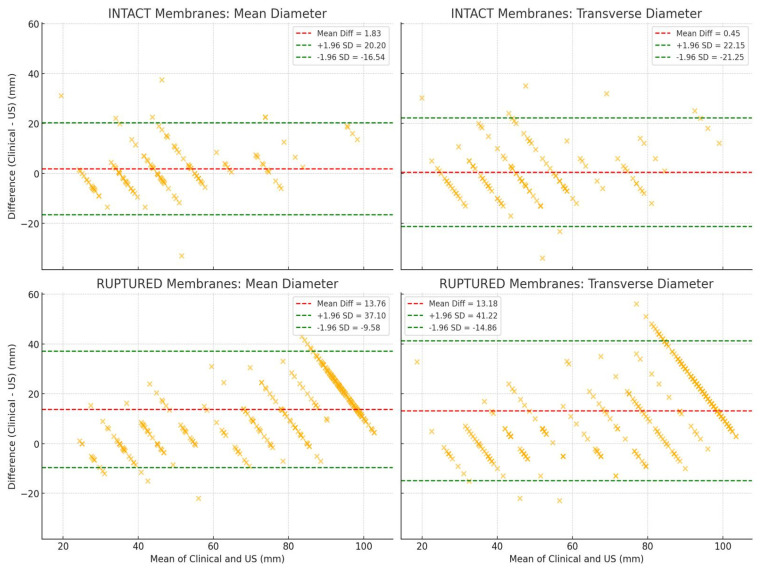
Bland-Altman plot analysis comparing clinical dilatation evaluation and mean ultrasound diameter of the dilatation calculated after ultrasound measurements and transverse diameter, divided by membrane status (the red dashed line shows the mean difference; the green dashed lines represent the limits of agreement—mean ± 1.96 × SD).

**Table 1 healthcare-13-01322-t001:** Characteristics of the study population.

Characteristic	Value Average (Min–Max, Median) Total—239 Patients
**Age (years old)**	26 (14–43, 24)
**BMI**	28 (18–40, 27)
**Living environment**	
Urban	107 pts. (44.77%)
Rural	132 pts. (55.23%)
**Pregnancy follow-up**	
Yes	149 pts. (62.34%)
No	90 pts. (37.66%)
**Gravidity**	3 (1–25, 3)
**Parity**	2 (1–8, 2)
**Gestational age by LMP/first trimester ultrasound (weeks)**	38 (30–41, 38)
**Gestational age by ultrasound at admission (weeks)**	36 (30–40, 36)
**Membrane status at admission**	
**Intact**	120 pts. (50.21%)
**Ruptured**	119 pts. (49.79%)
Clear amniotic fluid	109 pts. (45.61%)
Meconium-stained	10 pts. (4.18%)
**Number of vaginal examinations during labor monitoring**	4 (2–6, 3)
**Number of vaginal examinations per hour**	1.00 (0.17–2.42, 1.00)

**Table 2 healthcare-13-01322-t002:** Fetal and neonatal characteristics.

Characteristic	Value Average (Min-Max, Median)
**Estimated fetal weight by ultrasound (grams)**	3062 (1465–4148, 3100)
**Neonatal weight at birth (grams)**	3057 (1430–4140, 3070)
**Difference between estimated fetal weight and neonatal weight**	197 g (max. underestimation 730 g, max. overestimation 632 g, median difference 20 g)
**Neonate gender**	
Male	110 pts. (46.03%)
Female	129 pts. (53.97%)
**Apgar score**	9 (1–9, 9)

**Table 3 healthcare-13-01322-t003:** Membrane status impact on the different methods of ultrasound cervical measurement parameters compared with clinical examination.

Dilatation	Number of Patients with Intact Membranes (% from Subgroup)	Number of Patients with Ruptured Membranes (% from Subgroup)	Ultrasound Parameter—Dilatation Measurement	*p*-Value
2–3 cm	18 (62.07%)	11 (37.93%)	Mean Diameter	0.6199
			Anteroposterior Diameter	0.6034
			Transverse Diameter	0.8568
3–4 cm	26 (49.05%)	27 (50.95%)	Mean Diameter	0.5099
			Anteroposterior Diameter	0.9290
			Transverse Diameter	0.3973
4–5 cm	30 (55.55%)	24 (44.45%)	Mean Diameter	0.3379
			Anteroposterior Diameter	0.7405
			Transverse Diameter	0.4640
5–6 cm	28 (59.57%)	19 (40.43%)	Mean Diameter	0.1554
			Anteroposterior Diameter	0.7040
			Transverse Diameter	0.0947
6–7 cm	7 (31.81%)	15 (68.19%)	Mean Diameter	0.3590
			Anteroposterior Diameter	0.5487
			Transverse Diameter	0.1478
7–8 cm	11 (32.35%)	23 (67.65%)	Mean Diameter	0.0224
			Anteroposterior Diameter	0.1301
			Transverse Diameter	0.0109
8–9 cm	5 (12.50%)	35 (87.50%)	Mean Diameter	0.4864
			Anteroposterior Diameter	0.8533
			Transverse Diameter	0.4131
9–10 cm	0 (0%)	13 (100%)	Mean Diameter	n/a
			Anteroposterior Diameter	n/a
			Transverse Diameter	n/a
FULL DILATATION	4 (2.15%)	182 (97.85%)	Mean Diameter	0.1747
			Anteroposterior Diameter	0.1243
			Transverse Diameter	0.7925

**Table 4 healthcare-13-01322-t004:** Most accurate diameter measured by ultrasound in comparison with clinical examination for each dilatation level depending on membrane integrity.

Dilatation	Mean Difference Between Measurements Performed on Intact Membranes vs. Ruptured Membranes			Most Accurate Diameter Measurement by Dilatation Level
	Mean Diameter (mm)	Anteroposterior Diameter (mm)	Transverse Diameter (mm)	
2–3 cm	5.33	6.69	5.88	Mean Diameter
3–4 cm	5.26	6.70	7.28	Mean Diameter
4–5 cm	5.47	7.50	7.21	Mean Diameter
5–6 cm	7.20	9.04	9.20	Mean Diameter
6–7 cm	6.80	8.96	7.58	Mean Diameter
7–8 cm	7.38	10.47	9.06	Mean Diameter
8–9 cm	11.04	13.12	10.70	Transverse Diameter
9–10 cm	19.58	24.18	15.28	Transverse Diameter
FULL DILATATION	21.35	20.98	21.75	Anteroposterior Diameter

## Data Availability

The data that support the findings of this study are available by request from the corresponding author.

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
