# Peer review of "Membrane Status and Reliability of Intrapartum Transperineal Ultrasound in Cervical Dilatation Assessment"

_healthcare, 2025, doi:10.3390/healthcare13111322_

Round 1

Reviewer 1 Report

Comments and Suggestions for Authors

In this study the authors investigated the impact of membrane status on cervical dilatation evaluation through transperineal ultrasound compared to clinical examination and the degree of agreement between these two methods by dilatation level and membrane status. The intrapartum use of transperineal USG is a very current issue. There are some deficiencies in the article. I think the article can be re-evaluated after revision.

  1. How many clinicians performed USG and vaginal examination? What are the interobserver and intraobserver variability between them?
  2. Were vaginal examination and transperineal USG performed at the same time?
  3. How many of these pregnant women underwent labor induction?
  4. How was the number of cases determined?
  5. The limitations of the study should be added to the article.
  6. Does intrapartum transperineal USG have any advantages over vaginal examination? Why did the authors want to examine the cervical dilatation and the integrity of the membranes with this USG application? There is no explanation about these issues in the article.
  7. The authors should compare their findings with those in the existing literature and indicate how their article contributes to the literature.
  8. There are some studies in the literature on the determination of fetal vertex position by intrapartum perineal USG. The authors can also briefly mention this issue in their articles. Can both vertex position, cervical dilatation and membrane status be examined simultaneously by transperineal USG?

Author Response

Comments 1: How many clinicians performed USG and vaginal examination? What are the interobserver and intraobserver variability between them?

Response 1: Thank you for pointing this out. We agree with this comment. Therefore, we have added the information in the text, in the Materials and Methods part (in red). Also, since the evaluations were done by experienced physicians, we didn’t analyze the inter- and intra-observer variability, also considering that the evaluations weren’t performed simultaneously by two physicians and only just by one.

Comments 2: Were vaginal examination and transperineal USG performed at the same time?

Response 2: Yes, the vaginal and transperineal USG examinations were performed immediately one after the other, as mentioned in the Materials and Methods.

Comments 3: How many of these pregnant women underwent labor induction?

Response 3: None of them, as mentioned in the Materials and Methods, the induction of labor was an exclusion criterion.

Comments 4: How was the number of cases determined?

Response 4: The number of cases was not determined as a fixed value. We obtained the number of cases included in the study based on fixed length of the study, which was established from the beginning of the study (9 months).

Comments 5: The limitations of the study should be added to the article.

Response 5: Thank you for pointing this out. We included a new paragraph with all the limitations, at the end of Discussion chapter.

Comments 6: Does intrapartum transperineal USG have any advantages over vaginal examination? Why did the authors want to examine the cervical dilatation and the integrity of the membranes with this USG application? There is no explanation about these issues in the article.

Response 6: The intrapartum transperineal examinations have at least two main advantages, in comparison to vaginal examination: the reduced discomfort and the increases in objectivity of the evaluation. The objective of this study was not to evaluate the integrity of membranes by ultrasound examination.

Comments 7: The authors should compare their findings with those in the existing literature and indicate how their article contributes to the literature.

Response 7: Thank you for pointing this out. We added some more comparisons with the existing literature in the Discussion chapter.

Comments 8: There are some studies in the literature on the determination of fetal vertex position by intrapartum perineal USG. The authors can also briefly mention this issue in their articles. Can both vertex position, cervical dilatation and membrane status be examined simultaneously by transperineal USG?

Response 8: The evaluation of the fetal vertex or non-vertex position (deflected fetal head) was not the topic of this study. However, we have data on these aspects (evaluations performed transperineal and transabdominal to examine the degree of flexion or deflexion of the fetal head) and on the progression of the fetal head, but this data are due to be published in another paper.

Reviewer 2 Report

Comments and Suggestions for Authors

This manuscript addresses an important and underexplored aspect of intrapartum ultrasound—namely, the influence of membrane status on the accuracy of transperineal assessment of cervical dilatation—using a well-structured observational study design.

  • Language and Grammar: The manuscript contains numerous grammatical errors, awkward phrasing, and typographical mistakes (e.g., “it’s origins” instead of “its origins”; “intacte membranes” instead of “intact membranes”). A thorough English language editing by a native or professional editor is needed.

  • Scientific Clarity and Structure: The Introduction is informative but lacks focus. Key concepts and the rationale for the study should be presented more concisely and logically. Some historical references (e.g., from the 1970s) could be shortened or moved to a background section. The Discussion is repetitive in places and needs to better contextualize findings within current literature. Some speculation (e.g., on fetal head engagement) lacks citations or supporting evidence.

  • Statistical Reporting: While appropriate methods (e.g., Pearson correlation, Bland-Altman) are used, confidence intervals should be consistently reported, especially when interpreting effect sizes and differences. Bland-Altman plots and regression results are well presented but should be better integrated into the main discussion. The clinical relevance of the observed differences should be emphasized.

  • Terminology Consistency: Terms such as “mean diameter,” “transverse diameter,” and “anteroposterior diameter” should be defined clearly early in the text and used consistently. Occasional confusion arises due to overlapping or unclear usage.

  • Table and Figure Clarity: Some tables are dense and could benefit from clearer formatting and titles that directly indicate the variable of interest. Figures (e.g., ultrasound images and plots) lack legends or brief explanations. The figures would benefit from higher resolution and standardization of labeling.

  • Ethical and Methodological Transparency: Although ethical approval and consent are mentioned, the randomization method (if any) and how the sample size was determined are not clearly explained. It is not specified how inter-observer variability was addressed in clinical and ultrasound measurements (e.g., were results averaged between observers?).

  • Conclusions: The conclusion is valid but would benefit from a more precise clinical recommendation. For instance, stating how clinicians should modify practice based on membrane status and cervical dilation level.

Author Response

Comments 1: Language and Grammar: The manuscript contains numerous grammatical errors, awkward phrasing, and typographical mistakes (e.g., “it’s origins” instead of “its origins”; “intacte membranes” instead of “intact membranes”). A thorough English language editing by a native or professional editor is needed.

Response 1: Thank you for pointing this out. We agree with this comment. Therefore, we have performed a thorough analysis and grammar check of this manuscript and corrected all the mistakes.  

Comments 2: Scientific Clarity and Structure: The Introduction is informative but lacks focus. Key concepts and the rationale for the study should be presented more concisely and logically. Some historical references (e.g., from the 1970s) could be shortened or moved to a background section. The Discussion is repetitive in places and needs to better contextualize findings within current literature. Some speculation (e.g., on fetal head engagement) lacks citations or supporting evidence.

Response 2: We have done some modifications and corrections related to these aspects and we marked the corrections and additions with red in the text, for an improved clarity and for a better  Introduction and Discussion chapters.  Only one specific mention - there are no specific studies addressing the particular aspect that we studied in this study and for this reason the contextualization was hard to be done.

Comments 3: Statistical Reporting: While appropriate methods (e.g., Pearson correlation, Bland-Altman) are used, confidence intervals should be consistently reported, especially when interpreting effect sizes and differences. Bland-Altman plots and regression results are well presented but should be better integrated into the main discussion. The clinical relevance of the observed differences should be emphasized.

Response 3: We added the confidence intervals for Pearson correlation. For Bland-Altman we used the limits of agreement (upper and lower) in relation with the mean diameter. We emphasized the clinical relevance of the results in the Discussion chapter.

Comments 4: Terminology Consistency: Terms such as “mean diameter,” “transverse diameter,” and “anteroposterior diameter” should be defined clearly early in the text and used consistently. Occasional confusion arises due to overlapping or unclear usage.

Response 4: Agree. We have, accordingly, introduced the definitions of terms in the Materials and Methods and also corrected the inconsistencies in the rest of the article.

Comments 5: Table and Figure Clarity: Some tables are dense and could benefit from clearer formatting and titles that directly indicate the variable of interest. Figures (e.g., ultrasound images and plots) lack legends or brief explanations. The figures would benefit from higher resolution and standardization of labeling.

Response 5: We reduced the size of some tables and made the titles clearer. All the figures have a brief explanation, supplementary from the body of the article, the rest of the clarifications and explanations for the figures are found in the accompanying text. For the resolution of the figures - this was the maximum resolution for the figures to fit in the maximum size requested by MDPI.

Comments 6: Ethical and Methodological Transparency: Although ethical approval and consent are mentioned, the randomization method (if any) and how the sample size was determined are not clearly explained. It is not specified how inter-observer variability was addressed in clinical and ultrasound measurements (e.g., were results averaged between observers?).

Response 6: The patients were examined by the physician the admitted the case or by the on-call doctor (one of the 8 experienced physicians that performed the clinical examinations) and the ultrasound examination was done by one of the 7 specifically trained physicians, different from the previous ones, that was at the hospital or on call during the labor and delivery of a specific patient - this aspects were mentioned in a more scientific manner in the Materials and Methods. We didn’t use any randomization; we only blinded the vaginal examiner from the ultrasonographist for each case. If we would have done a randomization, because we analyzed spontaneous labors, this would have reduced dramatically the number of patients included in the study in order respect the randomizations process. The sample size was determined by time constraints - the length of the study was established from the beginning due to some internal factors specific to the Clinic in which the study was conducted. We included this aspect in the new limitations paragraph in the Discussion chapter. Also, the inter- and intraobserver variability was not a point of analysis for our study because we used blinded evaluations for each case, as mentioned before. So, yes, the results were averaged between observers, considering the sample size of this study - a larger one compared with other similar studies.

Comments 7: Conclusions: The conclusion is valid but would benefit from a more precise clinical recommendation. For instance, stating how clinicians should modify practice based on membrane status and cervical dilation level.

Response 7: Thank you for pointing this out. We have made some additions and modifications to the conclusion, to address this aspect that you mentioned.

Reviewer 3 Report

Comments and Suggestions for Authors

I commend the authors for this interesting investigation presenting the influence of membrane status on US evaluation of cervical dilatation during labour. I think some improvements can be made to better define the relationship of this manuscript with the existing literature.

  1. The repeated examinations increase the risk of infection, especially with membrane rupture. Please expand the discussion about this issue.
  2. Even though the accuracy of US technique is lower with advanced dilatations, this evaluation may be combined with head progression in those stages, especially in multiparous women. Distinction between nulliparous and pluriparous women is missing and therefore is a limitation of the study, but it can be investigated in future research together with the combination of angle of progression or head perineum distance for advanced dilatations. The final clinical scope is to understand the progress of labour.
  3. Please dedicate a space to strengths and limitations
Comments on the Quality of English Language

Thanks for this opportunity.

Some indications to improve the manuscript have been provided.

Best regards 

Author Response

Comments for Quality of English: The English could be improved to more clearly express the research.

Response: Agree. We have, accordingly, revised the manuscript and corrected all the errors of spelling or grammar and the inconsistencies.

Comments 1: The repeated examinations increase the risk of infection, especially with membrane rupture. Please expand the discussion about this issue.

Response 1: This aspect wasn’t analyzed in our study and we can’t expand further because we didn’t collect any data in this direction.

Comments 2: Even though the accuracy of US technique is lower with advanced dilatations, this evaluation may be combined with head progression in those stages, especially in multiparous women. Distinction between nulliparous and pluriparous women is missing and therefore is a limitation of the study, but it can be investigated in future research together with the combination of angle of progression or head perineum distance for advanced dilatations. The final clinical scope is to understand the progress of labour.

Response 2: We addressed the aspect of nullipara/multipara in the limitations paragraph that we newly introduced at the end of Discussion chapter. We have data on the head progression in relation to dilatation but we decided to not included in the article because would have done the article too complex to follow.

Comments 3: Please dedicate a space to strengths and limitations

Response 3: We addressed this recommendation and included a “Limitations” paragraph at the end of the Discussion chapter and also expanded the discussions and the conclusion.  

Round 2

Reviewer 1 Report

Comments and Suggestions for Authors

The authors have provided the necessary clarifications regarding the issues I mentioned. I consider the final version of the manuscript to be acceptable for publication.

Author Response

Thank you for your response! We kindly appreciated your feedback. 

Reviewer 2 Report

Comments and Suggestions for Authors

In my opinion, this manuscript is ready for publication.

Author Response

(The authors gave the same response as above.)

Reviewer 3 Report

Comments and Suggestions for Authors

I commend the authors for providing a revised version of their manuscript, that shows improvements according to the reviewers' suggestions. However, some additional observations deserve to be mentioned. 

1. I know that the risk of infection during vaginal examination is out of the main scope of the article. However, if the authors mention that US cervical examination can reduce women discomfort, I think it is also worth mentioning that reducing the number of VE may reduce the risk of maternal and neonatal infection, as suggested by some previous articles. Among them, you can see these references. This may be consider an additional advantage that has been discussed in the literature. 

i) Lemma K, Berhane Y. Early onset neonatal sepsis and its associatited factors: a cross sectional study. BMC Pregnancy Childbirth. 2024 Sep 28;24(1):617. doi: 10.1186/s12884-024-06820-5. PMID: 39342103; PMCID: PMC11438148.

ii) Gluck O, Mizrachi Y, Ganer Herman H, Bar J, Kovo M, Weiner E. The correlation between the number of vaginal examinations during active labor and febrile morbidity, a retrospective cohort study. BMC Pregnancy Childbirth. 2020 Apr 25;20(1):246. doi: 10.1186/s12884-020-02925-9. PMID: 32334543; PMCID: PMC7183634.

2. The discussion can be optimized according to previous evidence regarding sono-partogram and US cervical evaluation. Some references are listed below. Although some relevant articles are cited, there is no mention of partograo-sonopartogram. This revision would ensure optimal placement of this article in relation to the existing literature. 

i) Hassan WA, Eggebø T, Ferguson M, Gillett A, Studd J, Pasupathy D, Lees CC. The sonopartogram: a novel method for recording progress of labor by ultrasound. Ultrasound Obstet Gynecol. 2014 Feb;43(2):189-94. doi: 10.1002/uog.13212. PMID: 24105734.

ii) Dimassi K, Hammami A. Agreement between digital vaginal examination and intrapartum ultrasound for labour monitoring. J Obstet Gynaecol. 2022 Jul;42(5):981-988. doi: 10.1080/01443615.2021.1980513. Epub 2021 Dec 16. PMID: 34913801.

iii) Moncrieff G, Gyte GM, Dahlen HG, Thomson G, Singata-Madliki M, Clegg A, Downe S. Routine vaginal examinations compared to other methods for assessing progress of labour to improve outcomes for women and babies at term. Cochrane Database Syst Rev. 2022 Mar 4;3(3):CD010088. doi: 10.1002/14651858.CD010088.pub3. PMID: 35244935; PMCID: PMC8896079.

Author Response

Comments 1: I know that the risk of infection during vaginal examination is out of the main scope of the article. However, if the authors mention that US cervical examination can reduce women discomfort, I think it is also worth mentioning that reducing the number of VE may reduce the risk of maternal and neonatal infection, as suggested by some previous articles. Among them, you can see these references. This may be considering an additional advantage that has been discussed in the literature.

Response 1: Thank you for pointing this out. We have, accordingly, revised the Discussion chapter to emphasize this point and added the references that you recommended.

Comments 2: The discussion can be optimized according to previous evidence regarding sono-partogram and US cervical evaluation. Some references are listed below. Although some relevant articles are cited, there is no mention of partogram-sonopartogram. This revision would ensure optimal placement of this article in relation to the existing literature.

Response 2: Thank you for pointing this out. Therefore, we have included in the first paragraph of the  Discussion chapter an extended phrase regarding the sonopartogram and its usefulness, the necessity for further studies before including it in the routine labor monitoring and also added the references that you recommended.

We kindly appreciate your feedback!